# Improving Water Management Education across the Latin America and Caribbean Region

**Steve W. Lyon [1,2,\*], Peter Goethals [3], Petra Schneider [4], Luis Dominguez-Granda [5],
Henrietta Hampel [6], Norris Lam [1], Indira Nolivos [5], Frido Reinstorf [4],
Raymundo C. Rodríguez Tejeda [4], Raúl F. Vázquez [6] and Long Ho [3]**

[1] Department of Physical Geography, Stockholm University, Stockholm 10691, Sweden;
norris.lam@natgeo.su.se

[2] School of Environment and Natural Resources, Ohio State University, Columbus, OH 43210, USA

[3] Department of Animal Sciences and Aquatic Ecology, University of Ghent, Ghent 9000, Belgium;
peter.goethals@ugent.be (P.G.); long.tuanho@ugent.be (L.H.)

[4] Department of Water, Environment, Construction and Safety, University of Applied Sciences
Magdeburg-Stendal 35976, Germany; petra.schneider@h2.de (P.S.); frido.reinstorf@hs-magdeburg.de (F.R.);
raymondo.rodriguez-tejeda@hs-magdeburg.de (R.C.R.T.)

[5] Centre of Water and Sustainable Development, Escuela Superior Politecnica del Litoral,
ESPOL, Guayaquil 090150, Ecuador; ldomingu@espol.edu.ec (L.D.-G.); inolivos@espol.edu.ec (I.N.)

[6] Laboratorio de Ecología Acuática, Departamento de Recursos Hídricos y Ciencias Ambientales,
Universidad de Cuenca, Cuenca 010107, Ecuador; hennihampel@gmail.com (H.H.);
raul.vazquezz@ucuenca.edu.ec (R.F.V.)

\* Correspondence: steve.lyon@natgeo.su.se

**Abstract:** Education can help secure inclusive and resilient development around water resources. However, it is difficult to provide the latest science to those managing water resources (both now and in the future). Collectively, we hypothesize that dissemination and promotion of scientific knowledge using students as central agents to transfer theoretical knowledge into practice is an efficient way to address this difficulty. In this study, we test this hypothesis in the Latin America and Caribbean (LAC) region as a representative case study region. First, we use a literature review to map a potential gap in research on education around water resources across the LAC region. We then review potential best practices to address this gap and to better translate water resources education techniques into the LAC region. Integral to these efforts is adopting students as agents for information transfer to help bridge the gap between the global state-of-the science and local water resources management. Our results highlight the need to establish a new standard of higher educational promoting exchange between countries as local populations are vulnerable to future shifts in climate at global scales and changes in land usage at regional scales. The new standard should include peer-to-peer mentoring achieved by jointly exchanging and training students and practitioners in water management techniques, increasing access to water data and pedagogic information across the region, and lowering administration roadblocks that prevent student exchange.

**Keywords:** water; management; education; Latin America and Caribbean Region; Cuba; Ecuador

## 1. Introduction

Water is fundamental for the economy and quality of life in every country of the world. However, this renewable resource in some regions is increasingly threatened by human activities (e.g., pollution, overexploitation) [1,2]. Such threats are exacerbated by climate change and land use changes, i.e., agricultural frontier expansion and water used to feed growing populations and satisfy

other urban needs [3,4]. Balancing these various needs for water resources under the uncertainty of climatic changes creates a wicked problem for the future. The solution to these issues will require connecting water management to our best scientific understanding [5].

The nature of this problem can be seen across the Latin America and Caribbean (LAC) region [6]. According to the latest Intergovernmental Panel on Climate Change (IPCC) report in 2014 [7], Latin America is expected to experience significant climate-driven impacts on social, economic, and environmental realms [8,9]. There is a clear warming trend anticipated in this region over the next century. The LAC region's current projections indicate more or less the same amount of rain in the future, yet changes in the seasonal distribution of that rain, which will cause longer periods of drought and increased flooding severity [10]. Moreover, the hydrological cycle in the LAC region is influenced by complex meteorological phenomena such as "El Niño" and "La Niña", which produce extensive damage with regard to the local-to-global economy [10] and bring about significant human losses in the affected countries [11]. How we manage water resources across the LAC region is rapidly evolving (as is true across much of the world). Considering Ecuador, for example, several hydroelectric power stations have recently started to function and more are under construction in an attempt to secure and diversify energy sources [12]. For Ecuador, the government has set policies to reduce the consumption of conventional fossil energy and the 2008 constitution explicitly stated that the government would promote the use of hydropower energy to meet the growing energy demand for the coming 40 years [13]. Under the support of the National Electrification Master Plan 2013–2022, a major boost of hydropower generation has been triggered, which will help Ecuador target one of the cleanest energy mixes in the world [13]. As of 2017, 117 hydropower dams in Ecuador were mentioned in the literature, with 40 dams already finished and in service (i.e., about 60% of total power generation) [14]. Furthermore, 10 dams are currently under construction and another 67 dams have been proposed by the local government [15,16]. On the contrary from the other end of the LAC spectrum, Cuba's installed hydropower capacity and other renewable energy sources constituted just 1% of the country's total power generation [17]. Regardless, without adequate controls and regulations on environmental flow allowance from hydropower dams, strong impacts on the river functioning and ecosystem recovery capacity can be expected. Looking to wastewater, contamination due to the lack of control of industrial and agricultural activities and lack of water treatment plants prior to river disposal further aggravates water-related problems. In Cuba, some basic legislation and regulation for wastewater treatment does exist while in other Caribbean countries (i.e., the Bahamas, Grenada, and Haiti) such regulations do not exist [18].

Clearly, there is a need for innovation in water resource management across the LAC region. These innovations should be accessible online and developed to aid in everyday activities. Yet how can we connect regulatory agencies and managers in LAC with global innovations? Despite the recognition of water across Sustainable Development Goals (Agenda 2030) [19] and across the LAC region [20–22], innovative management concepts such as water conservation and protection, environmental flow definition and allocation, and securing of good ecosystem functioning are rarely considered (let alone planned for) in legal regulations and decision making. This leaves local populations vulnerable to future shifts in climate at global scales and changes in land usage at regional scales. Even more significantly, when we look to future generations of water managers, these integral aspects of water resources are rarely included in our curricula for higher education around water management [23]. We must therefore focus on boosting academic curricula in the field of water resources management to include up-to-date scientific and technological knowledge aimed at increasing the local skills and expertise of young professionals [23].

Therefore, it seems that education can help secure inclusive and resilient development around water resources. However, it is difficult to inform those managing water resources with the latest science [24]. In this study, we propose using students as central agents for dissemination and promotion of scientific knowledge into practice to address this difficulty. The focus of the study is on the LAC region due to its global importance and relevance. With about 34% of the world's annual renewable

freshwater resources and one of the largest volumes of freshwater resources per person (about 86,600 L per day), the LAC region has the potential to be an exemplar for global water resource management [25]. Yet how does water resource education in the region stack up relative to other regions? The LAC is a good testbed for our hypothesis and a region ripe for innovation. In the present study, the LAC region is used to explore the existence of a potential gap in water management education. To achieve this goal, we conducted a bibliometric analysis for mapping out a potential gap in LAC water management education. We then draw upon experiences within the project Water Management and Climate Change in the Focus of International Master Programs (Watermas) funded by the Erasmus+ Program of the European Union as a case study.

## 2. Materials and Methods

While some aspects of the LAC region are unique, we feel that the recommendations outlined can be of use to others. To identify the potential gap, we use a literature review approach. Then we identify best practices to address a potential gap.

### 2.1. Mapping a Potential Water Education Gap through Literature Review

We used the Scopus database to conduct a bibliometric analysis for mapping out a potential gap in LAC water management education. Scopus contains a relatively large international abstract and citation collection of peer-reviewed scientific literature. Scopus currently indexes 22,800 titles (e.g., journals, magazines, reports) from more than 5000 international publishers [26]. For this study, the database was accessed on 16 September 2019 and reflects publications from the period 1980 to 2019. As such, all subsequent data and bibliometric analysis presented are derived from that access date.

Since publications on water resources management tend to be interdisciplinary and the field is rapidly evolving, we limited our search in the Scopus database through key word selection. The template of the search queries applied in Scopus was as follows: TITLE-ABS-KEY AND AFFILCOUNTRY(*). This template was used in two distinct approaches for searching the database. In the first approach, the TITLE-ABS-KEY field was filled with the terms "water" AND "management" and the AFFILCOUNTRY(*) field was filled with the name of 33 countries in the LAC region (see Supplementary Material). We also considered the search using AFFILCOUNTRY(*) set to (1) the United States of America and Canada to establish a simple North American (NA) region that excluded the LAC region; (2) the 28 member states of the European Union (EU) to establish a relevant EU region; and (3) blank to assess the total global publications. The goal of this first approach was to map the current state of research on water management across the regions. In the second approach, the TITLE-ABS-KEY field was filled with the terms "water" AND "management" AND "educat*" and the AFFILCOUNTRY(*) field was filled in using the aforementioned regions. The goal of the second approach was to map the current state of scientific research on education in water management.

For all searches, the bibliographies (not articles) were downloaded from the Scopus website in the open-source statistical software R [27] using the *rscopus* package [28]. We clustered publications for analysis either as belonging to the LAC region, the EU region, or the NA region for comparison. All types of publications were assessed for the following characteristics: publication language, research categories and outputs, source journal, and contributing countries. From the data, knowledge orientation and future trends of research on water resources management and education can be identified and analyzed.

### 2.2. Establishing Best Practices for Addressing a Potential Water Education Gap

The Watermas project seeks to develop and establish a new standard of higher educational and scientific knowledge exchange between European and Latin American countries. By leveraging existing Master's courses and programs in water management at the various partner universities in Europe and across the LAC region (Table 1), Watermas supports curriculum development in the field of water resource conservation and management from the perspective of climate change adaption and impact mitigation.

**Table 1.** Partners and countries included in the project Water Management and Climate Change in the Focus of International Master Programs (Watermas).

| Name of Partner University | Country |
| --- | --- |
| University of Applied Sciences Magdeburg-Stendal | Germany |
| University of Ghent | Belgium |
| Stockholm University | Sweden |
| University of Cuenca | Ecuador |
| Escuela Superior Politécnica del Litoral (ESPOL) | Ecuador |
| University of Holguin | Cuba |
| University of Camagüey "Ignacio Agramonte Loynaz" | Cuba |

With an aim to enable the development of strategies for the adaptation of local water management facilities with regards to future challenges targeting a society-education-research nexus, the Watermas project utilized actions around mobilities to exchange, disseminate, and promote scientific knowledge and characteristics among the partner countries. Furthermore, the project developed actions around the technical and methodological training of students and educators involved in the countries concerned to strengthen their competencies and skills. The core of the Watermas project's theory of change lays in the exchange of teachers, researchers, and students from the different levels of higher education in different host universities and the creation of various teaching materials for training and study.

We implemented three main methods to synthesize the information relevant for this study. First, we used the quality plan developed for reporting project milestone achieved within the project. This quality plan was updated twice in the project's two-year lifespan, namely, at the end of year one and the end of year two. We explored changes in the rate of achieving milestones in this project between year one and year two relative to the roadblocks encountered. The quality report, thus, provides a rubric for metrics of success and growth in the Watermas project. Second, we synthesized the content of the travel reports written by the students participating in the project. These reports provided experiences and comments from student exchanges that helps qualitatively map out possible practices for closing education gaps as they are experienced and valued from the student perspective. Finally, we drew upon the experience of the educators and experts on both the European and LAC side of the project as sources of perspective for what worked and what did not work in practice. These experiences are invaluable as they reflect the reality of working with an education gap. Altogether, while acknowledging information anecdotal in nature, these lines of evidence and approaches converge around characterizing best practices to address a potential education gap across the LAC region.

## 3. Results and Discussion

### 3.1. What Does a Potential Gap in Water Education Look Like?

Our literature search returned 325,036 total publications on "water" and "management" with 16,051 for the Latin America and Caribbean (LAC) region, 92,879 for the European Union (EU) region countries, and 96,130 for the North America (NA) region. However, only a marginal proportion of the publications (about 2%) focused on education. Specifically, we found 7532 total publications on "water" and "management" and "education" with 350 for the LAC region, 1612 for the EU region, and 2855 for the NA region.

Publications on "water" and "management" and "education" were found in the following journals (listing journals with more than 100 publications): *Water Science and Technology; Science of the Total Environment; Journal of Environmental Management; Environmental Science and Pollution Research; IRRIGA; Revista Brasileira de Ciencia do Solo; Revista Brasileira de Engenharia Agricola e Ambiental; Chemosphere; Journal of Hazardous Materials; Water; Engenharia Agricola; Bioresource Technology; Agricultural Water Management; Environmental Technology* (United Kingdom); *PLoS ONE;* and *Tecnologia y Ciencias del Agua.*

For all publications including "water" and "management" and "education", about 98% of the publications were published in English, 1% in Spanish, and the rest in Portuguese or a mixture of the mentioned languages including French. This skew in publication language already starts to outline what a potential gap in water management education could look like.

### 3.1.1. Research in Water Management Versus Research in Water Management Education

Despite an increased scientific interest in water management education in LAC, which is indicated by the increase in publications by almost four times within the last 10 years (Figure 1a), there is still a substantial distance between LAC relative to the EU region and the NA region. The LAC region only represented 5% of all publications focusing on global water management education while just two countries in NA accounted for 38% and the EU region accounted for 21% (Figure 1b). This LAC scientific output in education related to water management can be held in contrast to the rich water resources in LAC countries (e.g., 34% of the world annual renewable freshwater resources). Further, there is considerable interest in water management across the LAC region as evidenced by the increasing scientific output of the region relative to that of the NA region considered. Specifically, the LAC region accounted again for about 5% of the global scientific output while the two NA countries accounted for about 30% and the EU region accounted for 28% of the global output (Figure 1c). Although these results show an increased interest in water management education across the LAC region, the output is still disproportionate relative to the scientific interest in water management of the region. Such a disproportionate investment in education while the science develops in the region maps out a potential education gap that could disconnect the research community's current state-of-the-science understanding from the knowledge base of the next generative of water managers.

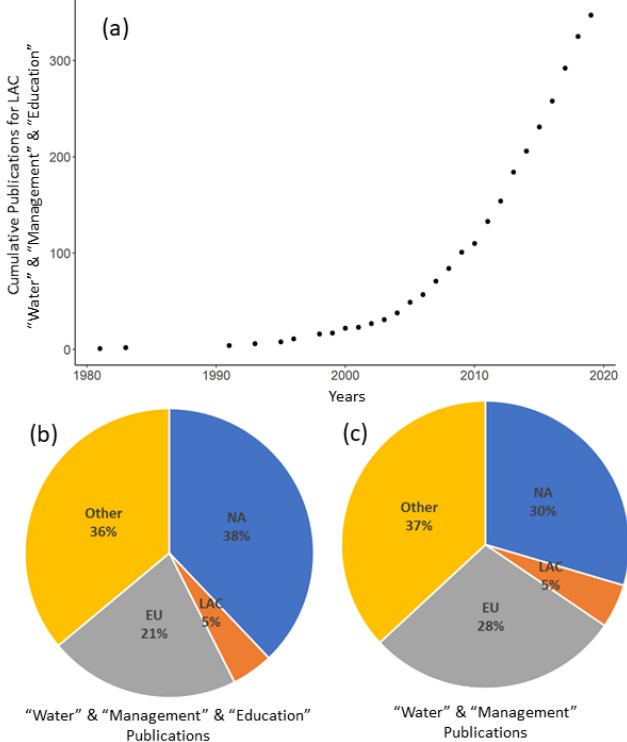

**Figure 1.** There has been an increasing output of publications for the Latin America and Caribbean (LAC) region on "water" and "management" and "education" over recent years (**a**). This increasing interest is despite a low proportion of the total global output as seen by the distribution of total publications with key words "water" and "management" and "education" (**b**) relative to "water" and "management" (**c**) in the LAC region, the European Union (EU) region, and the North America (NA) region from 1980 to 2019.

3.1.2. Word Clouds Comparing the Publication Keywords

From the top-used keywords, there is much commonality between the publications on water management versus those on water management education (Figure 2).

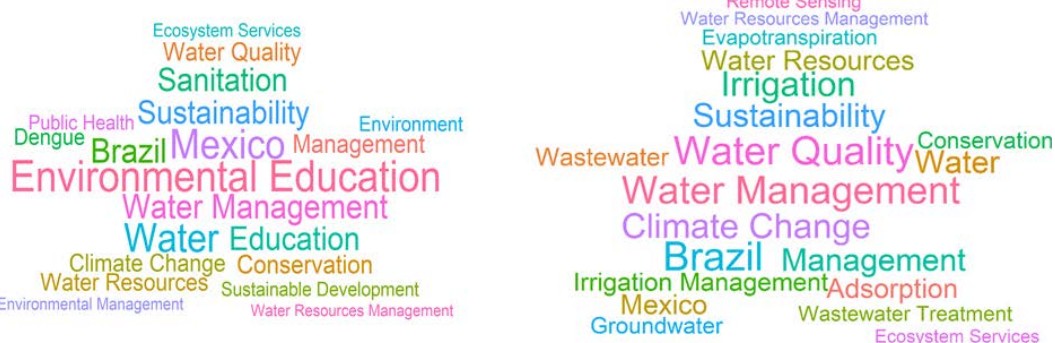

**Figure 2.** Word clouds of common key words for the Latin America and Caribbean (LAC) region's publications on "water" and "management" and "education" and "water" and "management" from 1980 to 2019 with the size of the word proportionate to the number of times it occurred.

As would be expected given our setup, and perhaps also indicative of a gap, the water management education word cloud references management but the water management word cloud does not reference education. More interestingly, besides general terms related to environment and water management, the water management word cloud reflects efforts around wastewater treatment and irrigation. These terms, however, are missing in the water management education word cloud. This demonstrates a disconnection between the current state-of-the-science and education in the LAC region. The emphasis on topics such as wastewater in research could also be a result of increasing concern over a severe shortage of wastewater facilities in LAC countries. This shortage has led to many outbreaks from waterborne diseases [29]. In fact, only about 20% of domestic wastewater was treated in the LAC region by 2008 [30] with the majority of treatment consisting of waste stabilization ponds [31]. The sustainable development of water resources is also another core focus in the water management research in LAC, which is indicated via the increasing recognition of water resources essential role in providing copious ecosystem services [32].

It should be noted Brazil and Mexico, which have the highest gross domestic products (GDPs) in the LAC region according to the World Economic Outlook database, appear as key words and are consistently the most productive countries, accounting for more than half of the publications of the whole region for both water management and water management education. On the other hand, Cuba and Ecuador, which were targeted in the Watermas project, were in the sixth and ninth position, respectively, for output in research on water management education among all LAC countries with only seven and three publications, respectively. There is apparent variability in the potential education gap such that it is not necessarily experienced by all countries across the LAC region in the same manner. This highlights the potential risk of not addressing an increasing education gap that could thereby "leave behind" countries in regions globally that are already vulnerable to water management risk.

*3.2. What Are Some Best Practices that Can Help Address a Water Education Gap?*

In the following, we characterize the best practices that were identified in addressing this education gap in the LAC region (Table 2). Support for these recommendations comes from the outcomes of the Watermas program and primarily through experiences in Ecuador and Cuba as case studies. In the following sections, we provide tangible examples as evidence for each best practice.

**Table 2.** Recommended best practices for closing an education gap in the Latin America and Caribbean (LAC) region based on experiences in the Water Management and Climate Change in the Focus of International Master Programs (Watermas) using Cuba and Ecuador as case studies.

| Recommended Best Practices for Closing the Education Gap in the Latin America and Caribbean (LAC) Region: |
| --- |
| • Peer-to-peer mentoring achieved by exchanging students |
| • Train students and practitioners jointly in water management techniques |
| • Increasing access to water data and pedagogic information across the region |
| • Lower administration roadblocks that prevent student exchange |

### 3.2.1. Peer-to-Peer Mentoring Achieved by Exchanging Students

Students appear to learn best directly from other students. This is consistent with problem-based learning concepts relevant for water management education [33] where problem scenarios tend to be open-ended and complex to encourage critical thinking, creativity, and peer-to-peer interaction [34]. Further, and directly addressing the connection between the state-of-the-science and management, pairing students can facilitate direct knowledge transfer peer-to-peer. An example of this from the Watermas project was partnering of students for sharing of novel approaches and technologies contributing to MSc thesis project work. Specifically, students from Stockholm University exchanged to Escuela Superior Politécnica del Litoral (ESPOL) working with Sentinel optical multispectral imagery and Synthetic-Aperture Radar (SAR), available from the European Space Agency's Copernicus program, to detect flood prone areas in Guayaquil, Ecuador. Within that exchange, the Swedish students partnered and worked directly with Ecuadorean educators and students at ESPOL. In these partnering pairs, the Swedish students were "forced" to explain their projects and explain how the remote sensing analysis could support or inform flood modeling efforts that the Ecuadorian students were leading. This is a direct knowledge transfer made peer-to-peer. Further, information exchange was not only one directional. The Ecuadorian students also had to explain the flood work they were doing in Guayaquil, which helped teach the Swedish students about the hydraulics of the region. This also gave the Ecuadorian students firsthand experience on translating their research into practical information to an invested audience, which is the cornerstone of stakeholder engagement in water management.

The concept of peer-to-peer mentoring also applies to PhD students as they begin their career as educators. For example, within the Watermas project the University of Cuenca exchanged a PhD student, who is also participating as a teacher in the University of Cuenca's MSc program "Engineering for the Management of Water Resources", to the University of Ghent for training on advanced and multivariate statistics techniques. Through this training and exchange, the student could incorporate both the new technical skills and the learning experience into development of teaching materials and approaches. This collectively contributes to a better education for the current MSc students at University of Cuenca on water management related issues. This "train-the-trainers" approach is also an important and perhaps more sustainable method to consider.

### 3.2.2. Train Students and Practitioners Jointly in Water Management Techniques

We experienced that cross disciplinary perspectives are needed to break down solos to solve societal water issues across the LAC region. This could be achieved in connection with practical training in experimental design and technical approaches to water management practitioners while promoting research and education for students. An example comes from the interactions between University of Ghent and University of Cuenca. Specifically, this interaction considers the discharge

from sewers and wastewater treatment plants (WWTPs), which is composed of high concentration of organic matter and nutrients. WWTP discharges can cause extensive modifications in the biochemical reactions in river systems, leading to an acceleration of greenhouse gas (GHG) emissions. However, the impacts of these discharges on GHG emissions from urban water systems (UWSs) are barely known [35,36] nor is it understood how to best monitor and mitigate these discharge impacts. This is a special technical need in the case of Ecuador as this country is highly vulnerable to climate change's impacts given its geographical position and topography [7].

From this perspective, two research groups—AECO in University of Ghent and PROMAS in University of Cuenca—collaborated in the Watermas project to organize training activities combined with a joint investigation. One of the main purposes was to train local practitioners and MSc students to assess the mechanisms of the GHG emission using different monitoring techniques to obtain a holistic assessment in UWSs. Further, after receiving sufficient theoretical knowledge, the local practitioners and students had opportunities to participate in a hands-on monitoring campaign for two weeks. An integrated monitoring assessment including physiochemical, hydromorphological, and meteorological monitoring was conducted together with the measurement of GHG concentrations and fluxes from both the UWSs and the WWTPs in Cuenca. Via the theoretical and practical trainings, the participants obtained a multidisciplinary perspective on the emerging issue and the scientific approach using novel technology to tackle it. Further, by working together, students were able to see the current state of play with regards to water management directly from the practitioners. This creates a win-win whereby students are empowered with information about current gaps in water management and with knowledge about how to address this gap drawing on international expertise.

Further examples for the joint training of students and practitioners arise from the collaboration of German students from the University of Applied Sciences Magdeburg-Stendal working at the Centre for Water and Sustainable Development (CADS) at ESPOL together with representatives of Durán Municipality. The students became part of the existing project ("Durán, Building a Resilience city: Strategies to Reduce the Vulnerability to Climate Change") that is performed by the climate change research group at ESPOL–GADM Durán. The scope of the students' work was the design of sustainable urban drainage systems (SuDS, [37,38]) for two municipal districts in Durán as a project counterpart.

After Woods-Ballard et al. (2007) [39], surface drainage systems developed in agreement with the ideals of the sustainable development are considered SuDS. According to the manual provided by Kondratenko et al. (2013) [40], the basic idea of SuDS is less about the implementation of a specific technique and more about a general design approach, which should: (1) be capable of managing precipitation runoff through several treatments; (2) have an ability to deal with rainwater runoff during extreme precipitation events; (3) be multifunctional by providing other functions (e.g., comfort functions, ecology) through rainwater management measures and adding water management functions to public space elements; and (4) have some cost efficiency and ease of service. The holistic nature of SuDS design, which aims to minimize the impact of developments on water resource quality and quantity and maximize the opportunities for amenities and biodiversity [39,41], is inherently complex making them a non-traditional approach for water managers. The student projects investigated an exemplarily SuDS design for the two districts of Durán working directly with local managers and experts from ESPOL. We consider this theory-practice integration [42,43] a substantial instrument to foster water management education across the LAC region.

### 3.2.3. Increasing Access to Water Data and Pedagogic Information across the Region

Data availability about natural and managed water resources were seen as a central limitation to education across the LAC region in the Watermas project. This is not necessarily a new observation as data limitations are a known problem for water resource management globally (e.g., [44,45]). What we have seen in our efforts for education development for Ecuador and Cuba is a lack of clear examples of relevant water resource management case studies as a major hinderance. The Watermas project tackled this roadblock by generating a database of projects to develop teaching modules and curriculum.

We have begun to populate this database with the relevant MSc thesis project work and accompanying datasets as case studies. For example, University of Cuenca generated information on the availability of hydro-meteorological data from local monitoring organizations in Ecuador. What was particularly relevant was the identification of a local enterprise (e.g., Municipal Public Enterprise for Drinking Water, Sanitation, and Telecommunications of Cuenca-ETAPA EP) responsible of collecting local hydro-meteorological data as the best potential data provider. Following from that effort, an official agreement was signed to get the monitored data archived and available for University of Cuenca and within our database for use in relevant case study development and MSc thesis projects. This creates a synergy between the thesis work currently being conducted by students and the curriculum developed for local academic programs. Further, and where possible, the Watermas database provides links to relevant national and global datasets to help facilitate their use in teaching activities.

In addition to populating the database with student thesis project output and links to data, we have also established and shared advanced pedagogic approaches that could help address the gap between the state-of-the-science and education across the LAC region. One example of such an advance pedagogic approach is recent work by Bring and Lyon (2019) [46] demonstrating role-play simulations as useful tools to achieve complex learning outcomes like those faced when educating around water management, i.e., making students able to coordinate and integrate various analytical skills in the complicated settings experienced in real-world water issues across sectors. To demonstrate this, the pedagogic approach looks at the effects of an integrated water resources management (IWRM) negotiation simulation evaluated next to more traditional teaching methods intended to foster quantitative understanding. Bring and Lyon (2019) [46] showed that despite similar student-reported achievement of both complex and quantitative intended learning outcomes, the students favored the negotiation simulation over the traditional method. This implies that role-play simulations can motivate and actively engage a classroom thereby creating a space for potential deeper learning and longer retention of knowledge, which would be advantageous across the LAC region.

Another useful pedagogic approach is the use of massive open online courses (MOOCs) to fill in where the curriculum lacks. Making these resources available across the LAC region can improve connectivity of students to global education resources. For example, providing a hydrology topic-related MOOC inventory [47] can serve as a clearing house of information. These MOOC resources can be at the basic level like those offered as a primer to understanding interactions between water and climate by TU Delft or at more advanced levels like those offered around irrigation innovations by University of Ghent. An added bonus of these MOOCs is the automatic translation option offered via internet resources such that content can be easily ported into Spanish. This reduces language barriers for students, which is a major roadblock across much of the LAC region.

### 3.2.4. Lower Administration Roadblocks that Prevent Student Exchange

Students often have the desire to exchange for education, but the roadblocks put in place by governments and universities are daunting. Some of these roadblocks are extremely challenging to overcome as they deal with infrastructure and costs. An example from our experience is that there is not affordable public or university housing available for students from the LAC region when trying to exchange to Stockholm University or to the University of Applied Sciences Magdeburg-Stendal. Other roadblocks, however, are more administrative and within our reach as academics to address and reduce. For example, we experienced many administrative barriers on both the European and LAC side with regards to establishing new bilateral agreements to allow student exchanges in the Watermas project. Counter to these new bilateral agreements, several universities in the Watermas project had existing bilateral agreements that made student exchanges easier from the first day of the project. We saw the impacts of this explicitly in our quality plan reporting out on the completion of student exchanges. At the end of year one of the Watermas project only universities with existing bilateral agreements were able to exchange MSc students. Those universities without pre-existing agreements took the entire first year to establish bilateral agreements. Once these agreements were established,

we saw that all universities could more easily send and receive students in the second year of the Watermas project. Clearly, if we are trying to connect students with the state-of-the-science we need to petition our faculties and universities to simplify the agreement processes to facilitate exchange.

*3.3. On the Need to Utilize Students as Vectors for Effective Knowledge Transfer*

Common across the best practices is the use of students as vectors for knowledge transfers and as catalysts for internationalization of existing education programs. Across the LAC region several Master's programs related to water resources management are currently offered with practitioners and technical staff from private and public (including governmental) institutions working for the water sector being among the target audience of these program. Exchanges of experiences and knowledge on water management via educational activities helped enrich program curricula of participant graduate programs from our efforts in Cuba and Ecuador (Table 2). In addition, student mobility for credit exchange has promoted the establishment and standardization of institutional policies at participant institutions. Altogether, these best practices provide trackable metrics (e.g., increased number and quality of student projects; development of aligned curricula; standardization of protocols and policies) of progress made in addressing and closing any potential LAC education gap. Further, as today's students become tomorrow's leaders, we would anticipate a shift in water management policy and practices from the status quo to more inclusive, holistic, and ultimately resilient approaches that weave together the tenets of sustainability.

From our literature review (Figure 1), the LAC region is currently not taking full advantage of a global emphasis on educating the next generation to ensure water resource sustainable development which is leaving the region vulnerable. Such a shortcoming creates a scenario where those potentially furthest "behind" are being left behind with regards to utilizing innovations from the realm of water management education. More effort on how we close this potential gap across LAC is needed since there may be cultural and financial differences impacting what works best. However, the best practices outlined here (Table 2) serve as a starting point. Further, the potential education gap around water management identified implies a need to strengthen of local capabilities to support informed decision making for a sustainable water use across the LAC region. The acquisition and training on equipment and modelling tools, such as was done in the Watermas project, should strengthen capabilities and allow a shift from current mostly descriptive studies towards more predictive approaches. This paradigm shift would enhance the confidence of management decisions from water authorities, where informed decisions based on scientific basis is crucial [48–50].

As referred to in UN Sustainable Development Goal number four, we must ensure inclusive and quality education for all and promote lifelong learning as quality education is the foundation for improving people´s lives and sustainable development. What is clear from this current study and the outcomes of the Watermas project is that the challenges faced by water managers in the LAC region demands a next generation of water professionals trained to provide innovative solutions and connect to the state-of-the-science. These young people are often discriminated because of age with regards to water management under the current paradigm. Through engagement of students in international exchange and empowerment with practitioners in their own countries, they begin to familiarize with characteristics of both scientific research and decision making in the water sector (e.g., stakeholder involvement for problem statement, collaborative multi-disciplinary research groups). It is expected that such students will have higher opportunities in the labor market, being even able to take influential positions in the water management sector of their home countries in few years.

## 4. Concluding Remarks

There is a clear need to improve the knowledge, skillsets, and competences of young leaders in the field of water management in the LAC region, especially so they can contribute to long-term growth, prosperity, and social inclusion. This improvement can only come about by incorporating scientific and technological knowledge into the curriculum development, as it will enhance capacities

to tackle future water resource management problem anticipated due to climate change. To achieve this incorporation in an efficient manner, we need to utilize students as the vector for knowledge transfer. This will ensure the increased local expertise needed to help optimize water use (i.e., water related ecosystem/environmental services) and sustainable exploitation of water resources is in the right hands. By enhancing the skills and competences of young people on water resource conservation and management, we can also ensure their competitiveness in the job market. The enhancement of young people, of course, also extends to early career educators (e.g., a train-the-trainers approach) that interact with our students. Altogether, by engaging students we can provide a straight-line to the next generation ensuring that "no one is left behind" on the road towards sustainable development to secure water for society.

**Supplementary Materials:** The following are available online at http://www.mdpi.com/2073-4441/11/11/2318/s1, Table S1: Latin America and the Caribbean (LAC) Table countries list.

**Author Contributions:** This article is based on the outcomes of the Watermas project. S.W.L. prepared the conceptualization of the manuscript and compiled the draft of the manuscript together with L.H., who along with support from P.G. performed the bibliographic analysis. P.S. contributed to the content of the manuscript, edited the layout, and contributed to the funding of the open access publication. L.D.-G., H.H., N.L., I.N., F.R., R.C.R.T., P.G. and R.F.V. revised and checked the manuscript. As project coordinator, F.R. was substantially responsible for the project funding acquisition.

**Funding:** This research was funded in part by the Erasmus+ Programme of the European Union (grant 586345-2017)).

**Acknowledgments:** This paper was prepared partially in the context of the project Water Management and Climate Change in the Focus of International Master Programs (Watermas) funded by the Erasmus+ Programme of the European Union. In this regard, this manuscript reflects only the views of the authors. As such, the European Commission's support for the production of this publication does not constitute an endorsement of the contents, which reflect the views only of the authors, and the Commission cannot be held responsible for any use which may be made of the information contained therein.

**Conflicts of Interest:** The authors declare no conflict of interest.

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
