# Peer review of "Improving Water Management Education across the Latin America and Caribbean Region"

_water, doi:10.3390/w11112318_

Round 1

Reviewer 1 Report

Lines 58-60 – add data on share (%) energy from hydropower in Ecuador’s total power generation.

Line 93-98 – eliminate repetition of ‘explore the existence of a potential gap in water management education’.

Chapter 2.1 – justify why you chose EU and NA regions for comparison and not other/more regions.

In lines 126-128 you state: ‘All types of publications were assessed for the following characteristics: document types and  languages, publication outputs, research categories, authors, journals, countries, institutions, and keywords.’ However, you don’t refer to document types, languages etc. in Results section, so information in lines 126-128 should includes only characteristics presented later in the text.

Line 231-234 – from ‘as such ..’ to ‘database’. it is too speculative for a scientific article; such a statement would require support from statistical analysis taking into account various factors; but in present form – maybe it’s true, maybe no, who knows.

Author Response

We wish to express our gratitude to all the reviewers that helped us to improve the manuscript. Hereafter we respond to each of reviewer comment (comment in italics with our response and edit made following directly after). We have also uploaded a revised document tracking all changes made to our manuscript as red text to allow for easy identification.

Reviewer #1

Lines 58-60 – add data on share (%) energy from hydropower in Ecuador’s total power generation.

Response: This was added in line 61, including a new reference

Line 93-98 – eliminate repetition of ‘explore the existence of a potential gap in water management education’.

Response: This was deleted in line 97-98

Chapter 2.1 – justify why you chose EU and NA regions for comparison and not other/more regions.

Response: We selected those regions as they can be considered as representative for the state of the art in water management education. We feel including more/other regions quickly creates a lengthy and open-ended basis for comparison.

In lines 126-128 you state: ‘All types of publications were assessed for the following characteristics: document types and languages, publication outputs, research categories, authors, journals, countries, institutions, and keywords.’ However, you don’t refer to document types, languages etc. in Results section, so information in lines 126-128 should includes only characteristics presented later in the text.

Response:  We have removed mention of the data not used in this study. We have edited the sentence to clarify:

“All types of publications were assessed for the following characteristics: publication language, research categories and outputs, source journal, and contributing countries.”

We added the representative journals in the also added information in the results on the language distribution informing on the potential education gap (line 186-189):

“For all publications including “water” and “management” and “education”, about 98% of the publications were published in English, 1% in Spanish, and the rest in Portuguese or a mixture of the mentioned languages including French. This skew in publication language already starts to outline what a potential gap in water education could look like.”

Line 231-234 – from ‘as such ..’ to ‘database’. it is too speculative for a scientific article; such a statement would require support from statistical analysis taking into account various factors; but in present form – maybe it’s true, maybe no, who knows.

Response: Agreed. We have removed the sentence in question. However, we have retained the statement of fact with regard to the relative GDP of Mexico and Brazil to the rest of the LAC in the opening sentence of the paragraph for regional context (line 238-241):

“Also noteworthy is that Brazil and Mexico, which have the highest gross domestic products (GDPs) in the LAC region according to the World Economic Outlook database, appear as key words and are consistently the most productive countries, accounting for more than half of the publications of the whole region for both water management and water management education.”

Reviewer 2 Report

I reviewed this paper and considered that it must be considered as a review paper instead of “article”. The authors based the investigation (very interesting!) on a review, I did not find any reasons to consider in another form. I attached a pdf with some specific comments. I consider that sometimes the authors write something informal adding questions, for example. I recommend you to adapt this language to another more adequated to the scientific literature. I would include a list of countries examined because not all the readers have to know the borders of LA. I would find necessary to include a list of the journal where you find the papers. Why? Because people can see where is more usual to find these kinds of papers. The paper is well-written, structured and very amazing. However, I consider that several parts need to be checked because you should include references. Why? Because it is very important to show the readers that the ideas were previously developed by other authors that you have under control!

Good luck and go ahead. I am looking forward to seeing the next version.

Author Response

We wish to express our gratitude to all the reviewers that helped us to improve the manuscript. Hereafter we respond to each of reviewer comment (comment in italics with our response and edit made following directly after). We have also uploaded a revised document tracking all changes made to our manuscript as red text to allow for easy identification.

Reviewer #2

I reviewed this paper and considered that it must be considered as a review paper instead of “article”. The authors based the investigation (very interesting!) on a review, I did not find any reasons to consider in another form.

Response: We thank you for your appreciation!

I attached a pdf with some specific comments.

Response: We added the responses to the pdf version (uploaded separately) and considered ourselves well served by those comments.

I consider that sometimes the authors write something informal adding questions, for example. I recommend you to adapt this language to another more adequated to the scientific literature. I would include a list of countries examined because not all the readers have to know the borders of LA.

Response: We have cleaned up any relaxed language throughout with our edits. Also, we have added the list of countries representing LAC as supplementary material – good suggestion.

I would find necessary to include a list of the journal where you find the papers. Why? Because people can see where is more usual to find these kinds of papers.

Response: We include the list of the journals containing the articles in the first part of our results (lines 179-185):

“Publications on “water” and “management” and “education” were found in following journals (listing journals with more than 100 publications): Water Science and Technology; Science of the Total Environment; Journal of Environmental Management; Environmental Science and Pollution Research; IRRIGA; Revista Brasileira de Ciencia do Solo; Revista Brasileira de Engenharia Agricola e Ambiental; Chemosphere; Journal of Hazardous Materials; Water; Engenharia Agricola; Bioresource Technology; Agricultural Water Management; Environmental Technology (United Kingdom); PLoS ONE; and Tecnologia y Ciencias del Agua.”

The paper is well-written, structured and very amazing. However, I consider that several parts need to be checked because you should include references. Why? Because it is very important to show the readers that the ideas were previously developed by other authors that you have under control!

Response: We have added several references particularly where missing and indicated in the annotated PDF file. Please see attached file for edits. We are glad the reviewer finds the paper well written and amazing!

Reviewer 3 Report

A very interesting paper about correlation between improvement in water resources management by improving the water management education and using students as "agents for information transfer to help bridge the gap between the global state-of-the science and local water resources management".

To be explained in the paper:

How will the improvement in water resources management and bridging the gap be "measured" in future (after the project has ended)?

What should be the next steps in research?

Author Response

We wish to express our gratitude to all the reviewers that helped us to improve the manuscript. Hereafter we respond to each of reviewer comment (comment in italics with our response and edit made following directly after). We have also uploaded a revised document tracking all changes made to our manuscript as red text to allow for easy identification.

Reviewer #3

A very interesting paper about correlation between improvement in water resources management by improving the water management education and using students as "agents for information transfer to help bridge the gap between the global state-of-the science and local water resources management".

Response: We are glad the review finds our effort interesting. Thank you for the kind words.

To be explained in the paper:

How will the improvement in water resources management and bridging the gap be "measured" in future (after the project has ended)?

Response: Throughout the manuscript, we highlight several metrics we considered for closing the education gap and how these translate to improvements in water management. These include exchanges of students, curriculum development and policy standardizations. We have added text to make the metric aspect of these outputs explicit (line 411-416):

“Altogether, these best practices provide trackable metrics (e.g. increased number and quality of student projects; development of aligned curricula; standardization of protocols and policies) of progress made in addressing and closing any potential LAC education gap. Further, as today’s students become tomorrow’s leaders, we would anticipate a shift in water management policy and practices from the status quo to more inclusive, holistic and ultimately resilient approaches that weaver together the tenets of sustainability. “

What should be the next steps in research?

Response: The next step is to continue to close the potential education gap across the region. Our best practices were derived from two partner countries. It is important to understand how these recommendations land and translate to other countries where cultural and financial barriers might be different. We added the following sentence to highlight this at line 421-423:

“More effort on how we close this potential gap across LAC is needed since there may be cultural and financial differences impacting what works best; however, the best practices outlined here (Table 2) serve as a starting point.”

Round 2

Reviewer 2 Report

Congratulations! The paper can be accepted!